# Bacteria That Made History: Detection of *Enterobacteriaceae* and Carbapenemases in the Waters of Southern Brazil’s Largest Flood

**DOI:** 10.3390/microorganisms13102365

**Published:** 2025-10-15

**Authors:** João Vitor Barboza Cardoso, Dariane Castro Pereira, William Latosinski Matos, Gabriela Simões de Oliveira, Victória Rodrigues de Carvalho, Louidi Lauer Albornoz, Afonso Luis Barth, Salatiel Wohlmuth da Silva, Andreza Francisco Martins

**Affiliations:** 1Laboratório de Pesquisa em Resistência Bacteriana, Hospital de Clínicas de Porto Alegre, Porto Alegre 90035-903, Brazil; jvcardoso@hcpa.edu.br (J.V.B.C.);; 2Laboratório de Microbiologia Molecular e Saúde Única, Universidade Federal do Rio Grande do Sul, Porto Alegre 90035-003, Brazil; 3Programa de Pós-Graduação em Ciências Médicas, Universidade Federal do Rio Grande do Sul, Porto Alegre 90035-003, Brazil; 4Programa de Pós-Graduação em Ciências Farmacêuticas, Universidade Federal do Rio Grande do Sul, Porto Alegre 90610-000, Brazil; 5Instituto de Pesquisas Hidráulicas, Universidade Federal do Rio Grande do Sul, Porto Alegre 91501-970, Brazil; 6Programa de Pós-Graduação em Recursos Hídricos e Saneamento Ambiental, Universidade Federal do Rio Grande do Sul, Porto Alegre 91501-970, Brazil

**Keywords:** antimicrobial resistance, carbapenemases, *Enterobacteriaceae*, Wastewater-based epidemiology, surveillance, molecular methods, flood, one health

## Abstract

Floods seriously threaten public health by promoting the spread of antimicrobial-resistant (AMR) bacteria, particularly in urban areas with poor sanitation. In May 2024, the state of Rio Grande do Sul, Brazil, experienced the most severe flood in its history, affecting over 2.3 million people and resulting in extensive dissemination of sewage, contaminating the environment. This study aimed to investigate the presence of *Enterobacteriaceae* and clinically relevant carbapenemase genes (*bla*_KPC_ and *bla*_NDM_) in floodwaters from Porto Alegre using molecular methods. Seventy-nine water samples were collected during four sampling campaigns conducted between May and June 2024. Samples were obtained from flooded areas and points across Guaíba Lake. DNA was extracted with the DNeasy PowerWater Kit, and qPCR was performed using TaqMan assays targeting *Enterobacteriaceae*, *bla*_KPC_ and *bla*_NDM_. Of the 79 samples, 75 yielded sufficient DNA for analysis. *Enterobacteriaceae* were detected in 100% of the samples, across all collections. The *bla*_KPC_ gene was detected in 100% of the first collection pools, and in 94.7%, 94.7%, and 85.7% of samples from the second, third, and fourth collections, respectively. The *bla*_NDM_ gene was present in 81.3% of the first collection pools, and in 78.9%, 89.4%, and 80.9% of samples from the subsequent collections. The high prevalence of *Enterobacteriaceae* and carbapenemase genes in floodwaters reveals an alarming environmental dissemination of AMR genetic markers. These findings underscore the need for environmental AMR surveillance, especially in disaster settings, and support the implementation of the One Health approach to mitigate the spread of resistance genes across human, animal, and environmental interfaces.

## 1. Introduction

Floods, among other common natural catastrophes, significantly impact human health by creating an environment favorable for pathogen proliferation, promoting the transmission of infectious diseases. In such scenarios, a comprehensive and coordinated public health response becomes essential to mitigate risks and safeguard affected populations [1].

In May 2024, the state of Rio Grande do Sul in south Brazil experienced the most devastating flood in its history, caused by the overflowing of rivers in the Guaíba Lake Hydrographic Basin. This catastrophe affected around 2.4 million people, resulting in 806 injuries, 184 fatalities and 25 people reported as missing [2]. As a consequence, urban sewage systems collapsed, leading to the backflow of both sanitary and stormwater sewage, contaminating homes, businesses, and public areas.

The urban sewage system acts as a large reservoir for bacteria, pharmaceuticals, and antimicrobial resistance genes (ARGs), making them a key source of epidemiological data. Hospital effluents, in particular, are rich in patient-derived bacteria, and may become hotspots for the emergence and dissemination of antibiotic-resistant pathogens [3]. This risk is exacerbated during flood events, when untreated effluents are widely dispersed into the environment.

It is important to highlight the serious challenges posed by Antimicrobial Resistance (AMR) in treating both hospital-acquired and community-acquired infections. The World Health Organization (WHO) identifies AMR as one of the top threats to global health and food security [4]. Bacterial antimicrobial resistance (AMR) was linked to an estimated 4.95 million deaths in 2019, surpassing fatalities from AIDS, HIV, and malaria. This makes AMR one of the leading global causes of death [5]. AMR poses a significant threat to the global economy, with the World Bank estimating in 2016 that, if not addressed, it could reduce the world’s annual Gross Domestic Product by 3.8% by 2050 due to loss of life and productivity. The economic consequences of AMR are expected to be particularly severe in low-income countries, potentially forcing an additional 28 million people into extreme poverty by the same year [6].

The WHO Bacterial Priority Pathogens List highlights bacterial species that demand urgent attention and investigation due to their potential to cause public health emergencies. This list categorizes pathogens based on transmissibility, mortality rates, outbreak potential, and the effectiveness of current treatments [7]. Among the pathogens included in our study are those classified as critical priority: *Enterobacteriaceae* and carbapenem resistance genes.

One of the most alarming manifestations of antibiotic resistance is carbapenem resistance, which is a growing concern in bacteria belonging to the *Enterobacteriaceae* group. This resistance is primarily attributed to the acquisition of carbapenemases. In fact, *Klebsiella pneumoniae* carbapenemase (KPC), a Class A carbapenemase, is frequently found not only in *Klebsiella pneumoniae* but also in other *Enterobacteriaceae*. However, it has also been identified in other Gram-negative pathogens, such as *Pseudomonas aeruginosa* [8]. Additionally, among the MBLs, the New Delhi metallo-β-lactamase (NDM) is one of the most frequently identified enzymes worldwide among *Enterobacterales* and *Pseudomonas aeruginosa* [8].

According to a multicenter study conducted in Brazil [9] carbapenem-resistant *Enterobacterales* (CRE) were the most frequent isolates in healthcare-associated bloodstream infections. They also found that the most prevalent carbapenemase genes in these isolates were *bla*_KPC_ and *bla*_NDM_, including the co-production of *bla*_KPC_ + *bla*_NDM_ [9]. Considering these findings, the aim of our study was to detect *Enterobacteriaceae* and carbapenemase genes directly in floodwater samples collected during and after the floods in Porto Alegre in 2024, using culture-independent molecular methods.

## 2. Materials and Methods

### 2.1. Water Sample Collection

For this study, a total of 79 water samples were analyzed. The samples were collected both during and after the May 2024 flood in Porto Alegre and its metropolitan region. In the initial sample phase, 73 samples were collected from different points in Guaíba Lake and from flood-affected urban areas (Figure 1). These samples were grouped into pools consisting of two or more samples from the same region, resulting in 17 composite samples (Table 1).

After the first collection, several flooded land points no longer existed due to water drainage. Therefore, in the second and third collections, a total of 20 samples (Table A1) were collected from 20 different points in Guaíba Lake (Figure 2). The previously collected points of Guaíba Lake remained the same; however, for flooded land points where water has drained, samples were collected from the Guaíba Lake at the shortest distance.

Finally, in the fourth collection, 22 water samples were collected (Figure 2). Of these, 20 corresponded to the same locations sampled in the previous two rounds, while 2 additional samples were included due to the appearance of two new flooded areas (Table A1) resulting from persistent rainfall.

**Figure 2 microorganisms-13-02365-f002:**
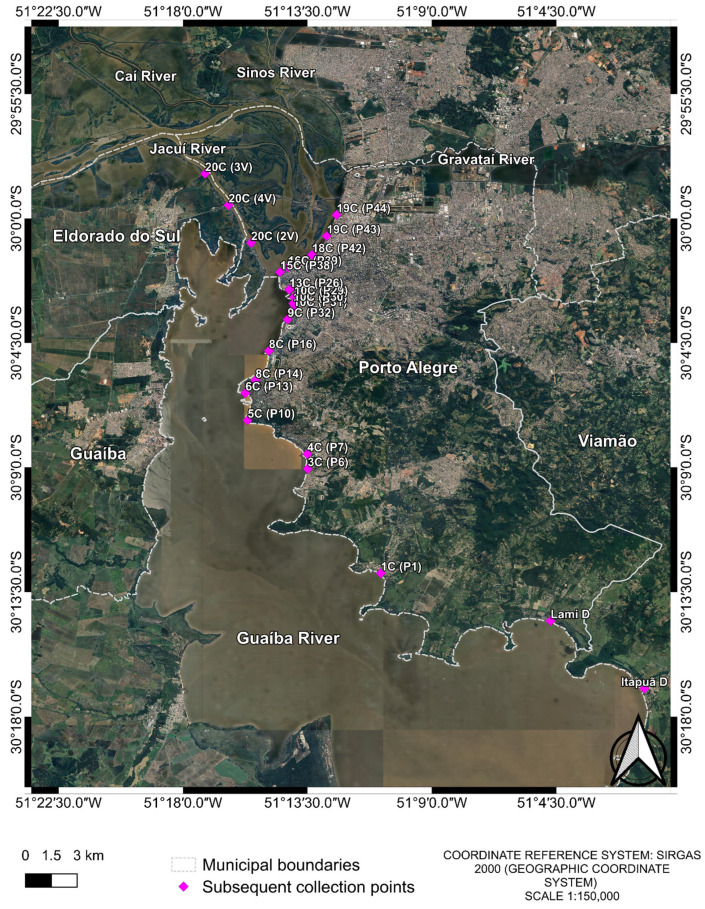
Collection points of the 2nd, 3rd and 4th collections. The two extra points from the 4th collection are “Lami D” and “Itapuã D”. Samples were named with a two-part code: the first part indicates the initial collection pool, and the second part denotes the subsequent collection points, e.g., First collection pool (Following collections). The image was produced by the authors using QGIS (version 3.40.5–Bratislava), with the satellite image obtained through the Google Satellite plugin. The coordinate reference system is SIRGAS 2000 (geographic coordinate system), and the representative scale is 1:150,000.

All samples were collected in duplicate, sub-superficially (approximately 10 cm below the waterline), in sterile 1 L bottles, and stored in a refrigerated thermal box [10]. A volume of 250 mL aliquots of each sample were sent for laboratory analysis.

### 2.2. Sample Transport

Environmental samples were transported from the collection site to the processing laboratory in a thermal box with ice. Samples were immediately stored at −80 °C until processing.

### 2.3. DNA Extraction from Water Samples

Samples were thawed and a volume of 100 mL was homogenized and filtered through a 0.45 µm membrane by vacuum filtration using sterilized glassware. After filtration, the membrane was submitted to total nucleic acid extraction using the DNeasy PowerWater Kit (Qiagen^®^, Hilden, Germany) according to the manufacturer’s instructions but using a volume of 35 µL instead of 100 µL for elution. DNA concentration and quality of the extracted material were assessed using Qubit™ 4 Fluorometer (Thermo Fisher Scientific™, Waltham, MA, USA) and NanoDrop™ Spectrophotometer (Thermo Fisher Scientific™, Waltham, MA, USA). All samples were stored at −80 °C until molecular analysis.

### 2.4. Detection of Enterobacteriaceae and Carbapenemases Genes by qPCR

Target detection was performed using a TaqMan Assay (Thermo Fisher Scientific™, Waltham, MA, USA) ID: Ba04329500_s1 for *Enterobacteriaceae* in a singleplex reaction. Detection of *bla*_NDM_ (ID: Ba04931076_s1) and *bla*_KPC_ (ID: Ba04646152_s1) genes, was carried out in a multiplex reaction. GoTaq^®^ Probe qPCR Master Mix (Promega^®^, Madison, WI, USA) reagent was used on the QuantStudio™ 3 Real-Time PCR System (Thermo Fisher Scientific™, Waltham, MA, USA). The thermal cycling conditions were as follows: 2′ at 95 °C for GoTaq^®^ DNA Polymerase activation followed by 40 cycles of 15″ at 95 °C for denaturation and 30″ at 60 °C annealing and extension. Samples that had been previously characterized by real-time PCR were utilized as positive and negative controls. Samples with Ct < 35 were considered positive.

## 3. Results

From the 79 initial samples a total of 75 did yield sufficient DNA extraction to proceed the qPCR reactions. *Enterobacteriaceae* was detected in 100% (75/75) of the samples (Figure 3). The *bla*_KPC_ gene was detected in 100% (16/16) of the pools from the first collection, with the *bla*_NDM_ gene detected in 81.25% (13/16) of these pools. Furthermore, the *bla*_KPC_ gene was found in 94.7% (18/19) of the samples from the second collection, 94.7% (18/19) of the samples from the third collection and 85.7% (18/21) of the samples from the fourth collection. The *bla*_NDM_ gene was detected in 78.9% (15/19) of the samples from the second collection, 89.4% (17/19) of the samples from the third collection and 80.9% (17/21) of the samples from the fourth collection. The Ct values for the samples are shown in Appendix A.

An interactive map has been developed to visualize the coordinates of each collection, providing information on the presence and absence of each target assessed in this study throughout the four collection periods (link for the interactive map: Interactive map (https://arturmattiaongaratto.shinyapps.io/Interactive_Map/).

## 4. Discussion

The widespread detection of *Enterobacteriaceae* and the high prevalence of the *bla*_KPC_ and *bla*_NDM_ genes in floodwaters following the May 2024 disaster in Porto Alegre highlights a critical environmental reservoir of antimicrobial resistance (AMR). These findings are particularly concerning given the direct exposure of communities to these contaminated waters, especially in urban areas where sanitation infrastructure was compromised.

Our results reinforce the recommendation from the WHO and the World Organisation for Animal Health (WOAH) to include *E. coli* and other members of the *Enterobacteriaceae* family in AMR surveillance and monitoring programs [11,12]. These organisms are not only ubiquitous in the environment and in the gastrointestinal tracts of humans and animals, but also exhibit high genetic plasticity, facilitating the horizontal transfer of resistance genes via mobile genetic elements (MGEs) [13,14]. The detection of *bla*_KPC_ and *bla*_NDM_ across multiple time points and locations suggests the persistence and dissemination of high-risk resistance determinants, likely amplified by selective pressures from human activity and environmental contamination [3,15,16]. In this context, the environment plays a pivotal role as both a reservoir and conduit for the spread of resistance. The floodwaters sampled in this study probably acted as vehicles for the redistribution of resistant bacteria and resistance genes across previously unconnected ecosystems, raising concerns about long-term ecological and public health impacts. That is why effective disaster response and mitigation strategies are crucial for reducing the health impacts of floods.

For research and monitoring of emerging and re-emerging pathogens to be effective, the use of molecular surveillance tools and technologies is essential. Natural disasters can create new opportunities for pathogen evolution, potentially leading to the emergence of more virulent or resistant strains. The spread of bacteria during floods poses a significant risk to public health, especially in areas with inadequate sanitation and limited access to potable water [1].

In this study, all target analytes were detected at sampling point 10C (P29) across all four sampling events. This site, located in the Dilúvio Stream delta, is significant because the stream receives untreated wastewater from major Porto Alegre hospitals along its banks, including Hospital A (890 hospital beds), Hospital B (394 hospital beds), and Hospital C (312 hospital beds). At point 11C (P19), sampled only during the initial flooding event, all three targets were also detected. This point is near Hospital D (312 hospital beds), which experienced inundation during the floods. At that time in the flood, the Guaíba Lake level was 5.09 m (Table A1), 1.49 m above flood stage. These results suggest the impact that the disposal of untreated hospital effluent can have on the environment and on the spread of multi-drug resistant pathogens.

The environmental dissemination of carbapenemase genes such as *bla*_KPC_ and *bla*_NDM_ also carries important therapeutic implications. These genes encode enzymes that confer resistance to carbapenems—antibiotics often used as a last resort in clinical settings. Their spread in aquatic environments, particularly following natural disasters, may contribute to the emergence of novel variants under selective pressure, complicating infection control efforts. The concept of WBE has already been applied to the monitoring of carbapenem-resistant *Enterobacterales* including environmental samples and treated wastewater [17,18,19] detecting different species carrying *bla*_NDM_, *bla*_KPC_ and *bla*_VIM_ [17,19,20].

Furthermore, within the context of wastewater surveillance, molecular tools have proven capable of monitoring genes at low abundances, aiding in the early detection of outbreaks involving established high-risk ARGs and ensuring data continuity for routine monitoring [21]. The occurrence of antimicrobial resistance genes in drinking water and urban sewage has already been evaluated previously in Porto Alegre. The presence of ARGs from different classes of antimicrobials was detected, such as carbapenemases, β-lactams, polymyxins, quinolones, sulfonamides, tetracyclines, and macrolides [16,22,23]. However, the sampling sites and methodology used were different from those of the current study.

This environmental reservoir can serve as a genetic source for reintroduction into clinically relevant pathogens, potentially undermining current treatment options [24]. Moreover, the continued expansion of such resistance mechanisms presents a significant challenge to the development of effective β-lactamase inhibitors. Medicinal chemistry efforts must now account for the growing diversity of resistance genes shaped not only by clinical antibiotic use but also by environmental and anthropogenic pressures [24,25]. Integrating environmental surveillance data into drug development strategies will be essential for anticipating resistance trends and designing more resilient therapeutic solutions.

Given these concerns, molecular tools such as real-time PCR offer a valuable and rapid means of environmental surveillance, particularly when culture-based approaches are not feasible. Wastewater-based epidemiology (WBE) has proven to be a powerful tool for community-level AMR monitoring [26,27,28,29,30,31,32,33]. In this study, WBE enabled the detection of resistance gene circulation in aquatic environments following natural disasters. Initially applied for pathogen surveillance, including SARS-CoV-2 [27] its use for AMR monitoring has expanded significantly in recent years, offering critical insights into the environmental dissemination of antimicrobial resistance genes (ARGs) and informing mitigation strategies [34].

These findings also underscore the relevance of the One Health framework, which recognizes the interconnectedness of human, animal, and environmental health, and is particularly relevant when assessing the impacts of floods. Floods create complex health challenges that demand integrated solutions. This holistic perspective not only enhances our ability to respond to immediate health threats but also strengthens the resilience of communities and ecosystems against future flooding [35,36,37]. The high frequency of resistance genes observed in this study points to the urgent need for integrated disaster preparedness plans that include microbiological monitoring and rapid response capacity.

Despite the valuable insights provided, our study has limitations. The use of molecular detection such as qPCR is suitable for rapid environmental surveillance; however, it does not allow for phenotypic confirmation or strain typing, or the assessment of bacterial viability. In particular, qPCR does not distinguish whether the detected resistance genes originate from viable bacteria or from free or degraded DNA. This is an important limitation when interpreting environmental AMR data, as it may overestimate the potential health risk even free DNA could be incorporated in the bacterial genome. Therefore, we emphasize the need for future studies to include culture-based approaches combined with phenotypic and genomic characterization of resistant strains. Such strategies would provide a more comprehensive understanding of AMR organisms circulating in post-disaster environments.

## 5. Conclusions

In conclusion, the May 2024 flood in Porto Alegre served as a stark reminder of how natural disasters can exacerbate existing public health threats. Environmental surveillance of antimicrobial resistance, particularly in vulnerable regions, is essential to inform timely interventions and protect community health.

## Figures and Tables

**Figure 1 microorganisms-13-02365-f001:**
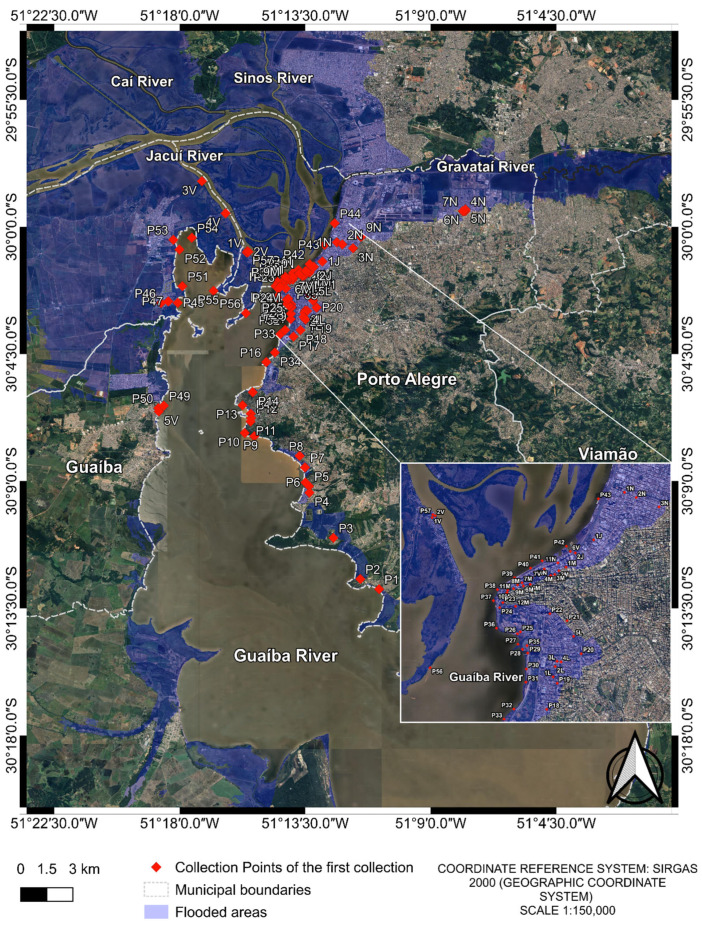
Collection points of the first collection. The image was produced by the authors using QGIS (version 3.40.5–Bratislava), with the satellite image obtained through the Google Satellite plugin. The coordinate reference system is SIRGAS 2000 (geographic coordinate system), and the representative scale is 1:150,000.

**Figure 3 microorganisms-13-02365-f003:**
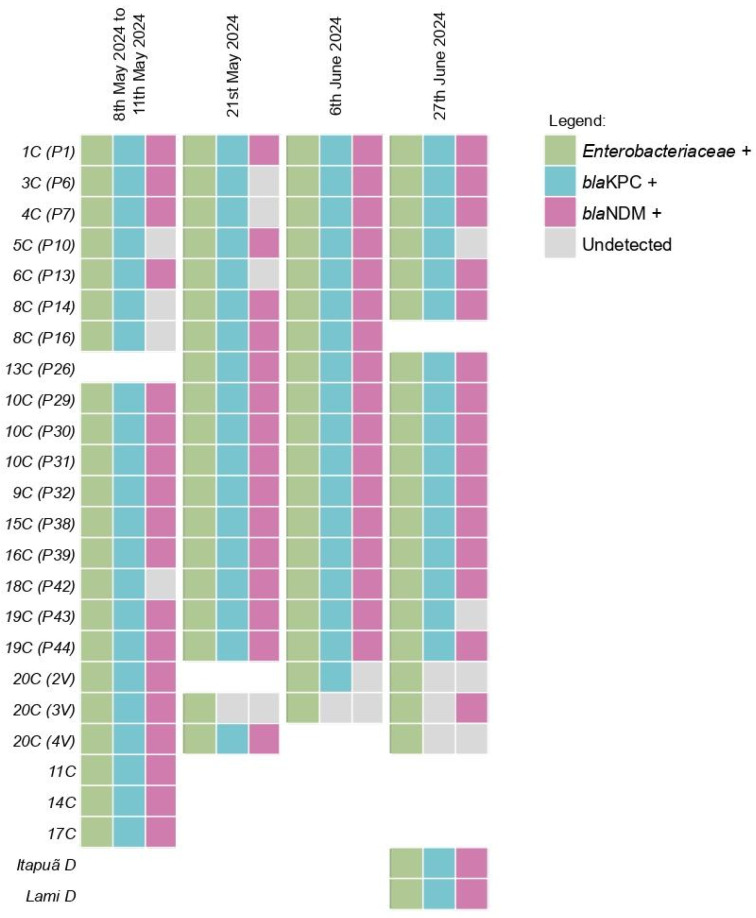
Detection profile of *Enterobacteriaceae* and Carbapenemases across the four collections in Guaíba Lake. Samples were named with a two-part code: the first part indicates the initial collection pool, and the second part denotes the subsequent collection points, e.g., first collection pool (following collections).

**Table 1 microorganisms-13-02365-t001:** Composition of the first collection’s pools.

Pool Identification	Collection Point
1C	**P1**, P2 e P3
3C	P4, P5 e **P6**
4C	**P7** e P8
5C	P9, **P10** e P11
6C	P12, P45 e **P13**
8C	**P14**, **P16** e P34
9C	P17, 18, **P32** e 33
10C	**P29**, **P30** e **P31**
11C	P19, 1L, 2L, 3L e 4L
13C	P28, P27, P35, P25 e **P26**
14C	P20, 5L, P21 e P22
15C	P36, P24, P37, P23 e **P38**
16C	10M, 11M, 9M, 8M, 6M, 5M, 7V, **P39** e P40
17C	P41, 10N, 4M, 3M, 2M, 11N e 1M
18C	**P42**, 6V, 2J e 1J
19C	**P43,** 1N, 2N, 3N, 9N e **P44**
20C	**3V**, **4V**, **2V** e 1V

Bolded are the collection points that were consistently collected in the following samplings.

## Data Availability

The original contributions presented in this study are included in the article/Appendix A. Further inquiries can be directed to the corresponding author.

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
