# Peer review of "Bacteria That Made History: Detection of *Enterobacteriaceae* and Carbapenemases in the Waters of Southern Brazil’s Largest Flood"

_microorganisms, 2025, doi:10.3390/microorganisms13102365_

Round 1

Reviewer 1 Report

Comments and Suggestions for Authors

The manuscript entitled "Bacteria That Made History: Detection of Enterobacteriaceae and Carbapenemases in the Waters of Southern Brazil's Largest Flood" presents an environmental surveillance study conducted after the historic flood that occurred in May 2024 in Porto Alegre, Brazil, where the presence of Enterobacteriaceae and carbapenem resistance genes (blaKPC and blaNDM) in floodwaters was assessed using molecular methods. This study highlights its innovative and timely approach within the One Health paradigm, which emphasizes the value of environmental surveillance as a key tool for anticipating and mitigating antimicrobial resistance threats following extreme weather events. Below are some recommendations to improve the quality of the manuscript:

  1. Although the manuscript adequately contextualizes the impact of the floods, it would be valuable to delve deeper into the clinical and epidemiological relevance of the environmental detection of blaKPC and blaNDM beyond the specific event. It is suggested to discuss how these findings could be linked to clinical outbreaks or surveillance policies, especially in nearby hospital settings.
  2. The use of qPCR is suitable for rapid molecular surveillance, but it does not distinguish whether the detected genes originate from viable bacteria or from free or degraded DNA. It is recommended, at least in the discussion, to acknowledge this limitation more emphatically and suggest future assays that include isolation and phenotypic and genomic characterization of resistant strains.
  3. Currently, the results only indicate the presence/absence of genes. Quantification of gene load (e.g., Ct or copies/mL) could significantly strengthen epidemiological and comparative analysis. If these data are available, could you be included? If not, this limitation should be noted in the discussion.
  4. Given that these are genes associated with carbapenem resistance, it would be pertinent to include a brief discussion of the therapeutic consequences of the environmental expansion of these enzymes (KPC and NDM), for example, their impact on the future development of new β-lactamase inhibitors, and the challenges facing medicinal chemistry in the face of environmental evolutionary pressure.
  5. The authors report detection frequencies by time and place, but do not explore associations with environmental variables (proximity to hospitals, rainfall intensity, type of water source, etc.). Although the primary objective is surveillance, including an exploratory model (e.g., logistic regression or spatial analysis) would enhance the interpretation.

Author Response

Dear Reviewers,

We sincerely appreciate the time and effort you have dedicated to reviewing our manuscript. Your insightful comments and suggestions have been invaluable in improving the quality and clarity of our work. In our response, all answers to the reviewers' comments are highlighted in blue, while all modifications made to the manuscript are marked in red for easy identification. We have carefully addressed each point raised and believe that the revisions have strengthened our manuscript. 

 Reviewer 1

1) Although the manuscript adequately contextualizes the impact of the floods, it would be valuable to delve deeper into the clinical and epidemiological relevance of the environmental detection of blaKPC and blaNDM beyond the specific event. It is suggested to discuss how these findings could be linked to clinical outbreaks or surveillance policies, especially in nearby hospital settings.

R: This important point has been addressed by expanding the discussion (L197-L215).

2) The use of qPCR is suitable for rapid molecular surveillance, but it does not distinguish whether the detected genes originate from viable bacteria or from free or degraded DNA. It is recommended, at least in the discussion, to acknowledge this limitation more emphatically and suggest future assays that include isolation and phenotypic and genomic characterization of resistant strains.

R: We agree that this is a limitation of qPCR-based detection in environmental samples. In response to your suggestion, we have revised the discussion section (L252–262) to more clearly acknowledge that qPCR cannot distinguish between DNA from viable organisms and non-viable or degraded material. We have also added a recommendation for future studies to include culture-based methods and phenotypic and genomic characterization of resistant strains to improve the interpretation of AMR surveillance data.

3) Currently, the results only indicate the presence/absence of genes. Quantification of gene load (e.g., Ct or copies/mL) could significantly strengthen epidemiological and comparative analysis. If these data are available, could you be included? If not, this limitation should be noted in the discussion.

R: Thank you for the suggestion! A table containing the Ct information for these samples has been included as Supplementary material (Table S1).

4) Given that these are genes associated with carbapenem resistance, it would be pertinent to include a brief discussion of the therapeutic consequences of the environmental expansion of these enzymes (KPC and NDM), for example, their impact on the future development of new β-lactamase inhibitors, and the challenges facing medicinal chemistry in the face of environmental evolutionary pressure.

R: Thank you for this excellent observation. This important point has been addressed by expanding the discussion (L225-L233) to include the therapeutic consequences of the environmental persistence of carbapenemase genes such as blaKPC and blaNDM. We now highlight how environmental reservoirs may serve as sources for novel or recombinant variants of resistance genes, potentially complicating treatment options and increasing the complexity of developing effective β-lactamase inhibitors. Additionally, the discussion emphasizes the challenges faced by medicinal chemistry due to evolutionary pressures occurring outside clinical settings.

5) The authors report detection frequencies by time and place, but do not explore associations with environmental variables (proximity to hospitals, rainfall intensity, type of water source, etc.). Although the primary objective is surveillance, including an exploratory model (e.g., logistic regression or spatial analysis) would enhance the interpretation.

R: In the discussion section (L197-206), we mentioned the relation among environmental variables (proximity to hospitals and rainfall intensity) and the presence of ARG. The suggestion to include an exploratory model is excellent, and we thank you for the contribution. However, as mentioned, this was not the primary objective of this study and may be included in future analyses. We believe the Interactive Map, which shows the distribution of biological markers across the city's collection points, is a good tool for data interpretation, similar to a spatial analysis.

Reviewer 2 Report

Comments and Suggestions for Authors

This study describes the possible effects of floods on the presence and level of certain AMR bacteria and genes in urban areas. This field is very important in terms of human health, since the number of AMR microbes is raising alarmingly.
The text is well written with good English, it is easy to read and understand.
The level of text similarity is not significant, self-citation (including authors and MDPI journals) is not an issue.

Line 52: For me, it sounds better that "approximately 2.4 million people". If such an exact number is given (2,398,255) I think "approximately" sounds a bit strange.

Line 293: Most of the journal titles are not abbreviated, so Trends. Anal. Chem. should be written non-abbreviated.

My questions:
1. As it is stated by the authors, the study has limitations, e.g. the temporal scope of sample collection was limited. I miss some preliminary data (published in other papers etc.) about the microbiological/AMR state of those selected sampling points that are available without flood (e.g. Dilúvio Stream delta or Guaíba Lake points). Do you have any information about the base level of bacterial contamination and AMR genes at these sites? What could be the extra load of bacteria and AMR genes during/after the flood?

2. Was the microbial contamination in the different hospitals' wastewater investigated previously? Are there any references about it? Are there other facilities (e.g. wastewater treatment plants) that might be the source of the detected contaminations and could be affected by the flood (e.g. Menino Deus - DMAE)?

3. The authors state "The use of molecular detection does not allow for phenotypic confirmation or strain typing" (line 216-217). Really, it would be informative to culture human pathogenic bacteria belonging to the Enterobacteriaceae family to see the proportion of carbapenemase harbouring and non-resistant isolates.
In addition, Acinetobacter baumannii and Pseudomonas aeruginosa may be reservoirs of the blaKPC and blaNDM genes, which may give sign during the qPCR reaction. Culturing may give some background information about these species (number, carbapenemase positive and negative isolates), as well.

4. The hospitals are mentioned in the text, but only one of them is localized (Hospital D near point 11C/P19). Could you mark the hospitals and the possible additional contamination sources (wastewater treatment plants) on the map?

Author Response

Dear Reviewers,

We sincerely appreciate the time and effort you have dedicated to reviewing our manuscript. Your insightful comments and suggestions have been invaluable in improving the quality and clarity of our work. In our response, all answers to the reviewers' comments are highlighted in blue, while all modifications made to the manuscript are marked in red for easy identification. We have carefully addressed each point raised and believe that the revisions have strengthened our manuscript.

Reviewer 2

Line 52: For me, it sounds better that "approximately 2.4 million people". If such an exact number is given (2,398,255) I think "approximately" sounds a bit strange.

R: We acknowledge that the use of “approximately” with a highly specific figure may appear inconsistent. Accordingly, we have revised the text to state “around 2.4 million people,” which we believe improves both the clarity and flow of the sentence. The sentence was changed as suggested.

Line 293: Most of the journal titles are not abbreviated, so Trends. Anal. Chem. should be written non-abbreviated.

R: We have corrected the journal title and now refer to it in its non-abbreviated form, Trends in Analytical Chemistry, in the revised document.

  1. As it is stated by the authors, the study has limitations, e.g. the temporal scope of sample collection was limited. I miss some preliminary data (published in other papers etc.) about the microbiological/AMR state of those selected sampling points that are available without flood (e.g. Dilúvio Stream delta or Guaíba Lake points). Do you have any information about the base level of bacterial contamination and AMR genes at these sites? What could be the extra load of bacteria and AMR genes during/after the flood?

R: Very well observed, thank you for the comment! However, we have no information about the baseline level of bacterial contamination and AMR genes at these sites. What we do have are data from a study (DOI: 10.1007/s42770-022-00786-2) that evaluated the occurrence of antimicrobial residues and antimicrobial resistance genes in drinking water treatment plants (DWTPs), sewage treatment plants (STPs), and sewage pumping plants (SPPs), though using a different method than ours. We added a topic to the discussion section (L216-224) addressing this issue.  It is important to note that this is the first study conducted at these specific points and analyzing these particular molecular markers. For future studies, it might be interesting to collect samples at the same collection points under normal conditions, without flooding, to make a comparison and obtain this data.

  1. Was the microbial contamination in the different hospitals' wastewater investigated previously? Are there any references about it? Are there other facilities (e.g. wastewater treatment plants) that might be the source of the detected contaminations and could be affected by the flood (e.g. Menino Deus - DMAE)?

R: There is a study published in 2012 (DOI 10.1007/s10482-012-9714-2) in which researchers analyzed effluent samples collected from four hospitals for Pseudomonas spp. and beta-lactamase genes. However, the researchers do not state which hospitals these are, so we cannot infer whether the wastewater from the hospitals we are referring to in this study has been previously investigated. We agree that Wastewater Treatment Plants are potential sources of contamination during floods. However, the Menino Deus - DMAE station, mentioned as an example, was not located within the flooded area, nor were 4 other DMAE stations (WTP Moinhos de Vento, WTP Belém Novo, WTP São João, and WTP Tristeza) (https://www.google.com/maps/d/u/0/viewer?mid=1yioyl2lr5xZOg3fV8i9OZV2t-5Jfa18&ll=-30.044433612782356%2C-51.22093287242908&z=14). Therefore, it is not considered a source of contamination detected in this study. Only the WTP Ilha da Pintada station, the smallest of them, which serves 5,800 inhabitants (https://lproweb.procempa.com.br/pmpa/prefpoa/dmae/default.php?reg=1&p_secao=172), was affected by the flood.

  1. The authors state "The use of molecular detection does not allow for phenotypic confirmation or strain typing" (line 216-217). Really, it would be informative to culture human pathogenic bacteria belonging to the Enterobacteriaceae family to see the proportion of carbapenemase harbouring and non-resistant isolates.
    In addition, Acinetobacter baumannii and Pseudomonas aeruginosa may be reservoirs of the blaKPC and blaNDM genes, which may give sign during the qPCR reaction. Culturing may give some background information about these species (number, carbapenemase positive and negative isolates), as well.

R: Thank you for your thoughtful and constructive feedback. We agree that culturing clinically relevant bacteria such as Enterobacteriaceae, Acinetobacter baumannii, and Pseudomonas aeruginosa would provide valuable complementary information to molecular data, particularly regarding the proportion of carbapenemase-producing versus non-resistant isolates. We also recognize that A. baumannii and P. aeruginosa can carry blaKPC and blaNDM genes and may have contributed to the qPCR signals observed in our study. In light of your suggestion, we have revised the discussion section to more clearly acknowledge this limitation and to recommend the inclusion of culture-based isolation and phenotypic/genotypic characterization in future studies. We believe this will help to clarify the species-level context of resistance gene dissemination in post-disaster environments.

  1. The hospitals are mentioned in the text, but only one of them is localized (Hospital D near point 11C/P19). Could you mark the hospitals and the possible additional contamination sources (wastewater treatment plants) on the map?

R: Thank you for your observation! The locations of hospitals A, B, C, and D have been added to the Interactive Map, along with the locations of the Menino Deus - DMAE and Moinhos de Vento - DMAE wastewater treatment plants (WTPs).

Round 2

Reviewer 1 Report

Comments and Suggestions for Authors

The authors took into account each of the recommendations provided and adjusted the manuscript to improve its quality. I believe the supplementary information in the document (Table S1) should be mentioned, and this supplementary information should include the title and authors of the manuscript. Best of luck!

Author Response

Dear Reviewer,

Thank you again for your contributions.

In response to your new requirement, we have mentioned Table S1 from the supplementary material in the main body of the manuscript (lines 157-158) and have also added the study's title and authors' names to the supplementary material.